# Molecular Mechanism and Prevention Strategy of Chemotherapy- and Radiotherapy-Induced Ovarian Damage

**DOI:** 10.3390/ijms22147484

**Published:** 2021-07-13

**Authors:** Seongmin Kim, Sung-Woo Kim, Soo-Jin Han, Sanghoon Lee, Hyun-Tae Park, Jae-Yun Song, Tak Kim

**Affiliations:** 1Gynecologic Cancer Center, CHA Ilsan Medical Center, CHA University College of Medicine, 1205 Jungang-ro, Ilsandong-gu, Goyang-si 10414, Korea; naiad515@gmail.com; 2Department of Obstetrics and Gynecology, Seoul National University Hospital, 101 Daehak-ro, Jongno-gu, Seoul 03080, Korea; byulbi@naver.com (S.-W.K.); skimgyn@chamc.co.kr (S.-J.H.); 3Department of Obstetrics and Gynecology, Korea University College of Medicine, 73 Inchon-ro, Seongbuk-gu, Seoul 02841, Korea; cyberpelvis@gmail.com (H.-T.P.); sjyuni105@gmail.com (J.-Y.S.); tkim@korea.ac.kr (T.K.)

**Keywords:** chemotherapy, radiotherapy, gonadotoxicity, fertility preservation, embryo cryopreservation, oocyte cryopreservation, ovarian tissue cryopreservation, oocyte in vitro maturation, ovarian suppression, oncofertility

## Abstract

Fertility preservation is an emerging discipline, which is of substantial clinical value in the care of young patients with cancer. Chemotherapy and radiation may induce ovarian damage in prepubertal girls and young women. Although many studies have explored the mechanisms implicated in ovarian toxicity during cancer treatment, its molecular pathophysiology is not fully understood. Chemotherapy may accelerate follicular apoptosis and follicle reservoir utilization and damage the ovarian stroma via multiple molecular reactions. Oxidative stress and the radiosensitivity of oocytes are the main causes of gonadal damage after radiation treatment. Fertility preservation options can be differentiated by patient age, desire for conception, treatment regimen, socioeconomic status, and treatment duration. This review will help highlight the importance of multidisciplinary oncofertility strategies for providing high-quality care to young female cancer patients.

## 1. Introduction

It is estimated that more than 9.2 million women were newly diagnosed with cancer worldwide in 2020 [1]. Furthermore, there were 89,500 new cancer cases and 9270 cancer deaths in adolescents and young adults (AYAs) aged 15–39 years in the United States [2]. The survival of cancer patients has significantly improved due to recent advances in cancer treatment [3,4]. However, oncologic therapies can affect ovarian function in young women [5,6,7,8]. The exhaustion of ovarian follicle reservoirs may lead to not only loss of fertility but also premature ovarian failure, which could result in poor quality of life in young female cancer survivors [9,10,11]. Recently, fertility preservation (FP) has become an emerging discipline with significant clinical value in the care of AYA cancer patients [12,13,14], and many organizations have provided recommendations for FP during cancer treatment [15,16,17,18,19].

Chemotherapy has toxic effects on the ovaries and causes the loss of the primordial follicle (PF) reserve [20]. Endocrine therapy can increase the risk of infertility in patients with hormone receptor-positive malignancies [21]. In the case of abdominal or pelvic cancers, treatments including radiotherapy or surgery may alter future fertility because of direct gonadal damage [22,23]. Many studies have explored the mechanisms implicated in ovarian toxicity during cancer treatment; however, the underlying molecular pathophysiology is not fully understood [24,25,26,27,28].

This article will review the mechanisms of cancer therapy-induced ovarian dysfunction and explore the future perspectives for preventing infertility in AYAs with cancer.

## 2. Regulation of the Ovarian Follicular Reserve

In the ovary, there is a finite number of PFs that continue to decline until menopause [29]. A single layer of granulosa cells (GCs) surrounds an immature oocyte to form each PF, which together constitute the “ovarian reserve”, which refers to the total population of PFs [30]. In mammals, the ovarian reserve is established early in life and then declines regularly throughout the reproductive years. In the human ovary, 85% of the potential oocytes are lost before birth [30]. PFs usually remain quiescent for years, and this quiescence is maintained by several molecules, including PTEN, the TSC1–TSC2 complex, Foxo3A, P27, anti-Müllerian hormone (AMH), and FoxL2 [31]. Most PFs then undergo atresia, but a minority of follicles reach the pre-ovulatory stage. The activation of dormant follicles is mediated by the upregulation of the PI3K/PTEN/Akt pathway [32,33,34,35]. The survival of PFs is regulated by PDK1 signaling or RPS6 [32]. Several studies have reported that the ovarian reserve is regulated by autophagy [36,37]. Bcl-2 and its associated proteins, such as BAX, play critical roles in the survival or apoptosis of PFs [32]. Therefore, a combination of follicular activation and multiple inhibitory/activator molecules is necessary to preserve the ovarian reserve, and any disturbance in these mechanisms may induce a premature loss of ovarian function [38].

## 3. Cancer Treatment-Induced Ovarian Damage

### 3.1. Mechanism of Chemotherapy-Induced Ovarian Damage

Chemotherapy can adversely affect the ovarian reserve [39,40,41,42,43,44,45]. Table 1 lists known gonadotoxic chemotherapy agents [29,45,46,47]. Chemotherapy-induced amenorrhea may be transient, and menstruation may recur after treatment completion. The oocytes and GCs are vulnerable to chemotherapeutic agents. Each agent may have a different mechanism of action on cancer cells, resulting in the cessation of the cell cycle.

#### 3.1.1. PF Loss via DNA Alteration, Follicular Atresia, and Apoptosis

Chemotherapeutic agents can induce double-stranded breaks (DSBs) in DNA. If a DSB is repaired successfully, the cell survives via ataxia–telangiectasia mutated (ATM)-mediated DNA damage repair pathways. However, if the repair pathways fail, DNA damage can result in cellular apoptosis [48] (Figure 1). Apoptosis in the ovary has been demonstrated in growing follicles and has been shown to originate in proliferating GCs [49]. Within mature oocytes, the P63 protein activates BAX and BAK proteins, which can be transmitted by the activation of Tap73, a P53-upregulated modulator of apoptosis, and phorbol-12-myristate-13-acetate-induced protein 1 [50]. Many studies have demonstrated these mechanisms in vitro and in vivo [51,52,53,54,55]. Notably, even chemotherapeutic agents with low gonadotoxicity can alter the development of growing follicles because mitotic cells are especially sensitive to chemotherapeutic agents [51,55].

#### 3.1.2. Follicle Loss via Activation and “Burnout”

The PI3K/PTEN/Akt pathway terminates follicle dormancy, directly influences the oocytes and pre-GCs of PFs, and indirectly destroys large follicles [24]. The destruction of follicles induces the impairment of AMH and reduces the suppression of the PF pool, which is followed by the activation of PFs in an attempt to compensate for the decrease in the number of growing follicles [56]; this phenomenon is called the “burnout effect” [57,58] (Figure 2). In a previous study, exposure to 3-methylcholanthrene, a carcinogen and ovotoxicant, resulted in PF activation and depletion [59], implying that the burnout effect triggers the growth of dormant follicles. However, this induction of dormant follicle growth is affected by the upregulation of the PI3K/PTEN/Akt pathway and substantial follicular apoptosis, which results in a reduction in AMH secretion. This could be caused by a direct effect on oocytes and pre-GCs [56,60,61]. Reduced AMH levels and enhanced follicular recruitment and atresia in patients with ovarian endometriosis may have the same effect [62]. However, the mechanism by which chemotherapeutic agents induce the activation of this pathway is still under investigation.

#### 3.1.3. Stromal and Microvascular Damage

The ovarian stroma can be indirectly damaged by chemotherapeutic agents [29,63]. For example, the administration of doxorubicin results in a significant reduction in ovarian blood supply and small vessel spasms in the ovary [64]. Another study reported that chemotherapy led to stromal fibrosis and ovarian vascular abnormalities [65]. These findings imply that damage to blood vessels and focal fibrosis of the ovarian cortex could be another mechanism of chemotherapy-induced ovarian dysfunction [66]. Additionally, in patients previously exposed to chemotherapy, the ovaries show thickening and hyalinization of cortical stromal vessels caused by the disorganization of blood vessels in the ovarian cortex and cortical fibrosis [67]. The production of sex steroids decreases because of damage to endocrine ovarian function after exposure to chemotherapeutic agents [68]. This is also supported by a recent study that showed an inverse correlation between ovarian vascular density and PF apoptosis [69], thus suggesting an indirect mechanism by which chemotherapy-induced ovarian vascular injury reduces the number of PFs.

### 3.2. Radiation-Induced Ovarian Damage

Ionizing radiation to the abdominopelvic region has deleterious effects on gonadal function at all ages [70]. For example, cervical and rectal cancers usually require pelvic irradiation, and craniospinal radiotherapy is performed in cases of central nervous system malignancy. In some patients with Hodgkin’s disease, pelvic lymph nodes require irradiation, and total body irradiation may be necessary prior to bone marrow transplantation.

The resulting damage depends on the dose and field of irradiation and the age of the patient. Women who received radiation treatment outside the pelvis had a low risk of ovarian dysfunction [71]. In the prepubertal period, the ovaries are relatively resistant to gonadotoxicity [72].

#### 3.2.1. Radiosensitivity of Oocytes

Dividing GCs appear to be the main target of radiation-related gonadotoxicity. Prominent cell death has been observed within a few hours of irradiation [73]. Oocytes are highly radiosensitive because the estimated dose at which half of the follicles are lost in humans (LD50) is <2 Gy [74]. A single oocyte is highly radiosensitive to a D_0_ of 0.12 Gy (reciprocal of the slope of the exponential region of a survival curve). This sensitivity is affected by age; women younger than 40 years of age are less sensitive, requiring 20 Gy to experience permanent damage, whereas older women require only 6 Gy [75]. The radiosensitivity of oocytes differs according to their growth phase. A quiescent PF is usually more radio-resistant than a large maturing follicle [74]. Radiotherapy-induced ovarian damage also occurs in the stroma with vascular damage, resulting in tissue atrophy and fibrosis [73]. In general, a combination of multiple factors determines the extent of radiosensitivity, including age, the use of combination therapy, and radiation dose [76].

#### 3.2.2. Linear Energy Transfer

The biological effect of radiation treatment is also affected by linear energy transfer (LET) in tumors [77]. LET radiation induces anticancer effects by depositing physical energy or radiation into malignant cells, which results in stable free radicals and induces cellular damage because of the direct ionization of the cellular macromolecules, such as DNA, RNA, lipids, and proteins [78]. High LET radiation results in gonadal DNA damage that causes multiple lesions within the helical turns of the DNA, which is referred to as “direct” damage (Figure 3).

#### 3.2.3. Oxidative Stress Resulting in DNA Damage

Radiotherapy also leads to the generation of reactive oxygen species (ROS) in cancer cells as a result of water radiolysis, inducing oxidative stress and the diminution of the antioxidant defense mechanisms, which could also affect healthy normal tissues, including the ovaries [79]. Thus, the imbalance between the free radicals and the oxidative radicals may play a role in the etiology of radiotherapy-induced gonadotoxicity [80]. Increased ROS levels induce rapid PF loss and female infertility via DNA damage, which is termed as “indirect” damage [81] (Figure 3). ROS can induce apoptosis, leading to oxidative stress exerted on cellular macromolecules, ultimately activating the intrinsic mitochondrial pathway of apoptosis via p53 activation, which results in cytochrome c release and caspase activation [82,83]. Activated caspases cleave DNA damage repair enzymes to block cellular DNA repair and enhance apoptosis [84]. Another study identified increased activity of the MAPK signaling pathway in irradiation-induced GC apoptosis [85]. Human oocytes express DNA repair genes, but their role in the repair of radiation-induced genomic damage is unclear [86]. As ionizing radiation generates free radicals that induce DNA damage, compounds that scavenge free radicals may be useful for protection against radiation damage [87].

## 4. Detection of Ovarian Damage

### 4.1. Clinical Considerations

Older age is an important risk factor for ovarian damage compared to younger age. A significantly higher incidence of amenorrhea after chemotherapy was observed in female patients with breast cancer aged >40 years than in younger women [88]. Another study confirmed similar results [89]. A cross-sectional analysis of data from a prospective cohort study demonstrated that the ovarian reserve was impaired in a dose-dependent manner in patients exposed to cancer therapies, including chemotherapy or pelvic radiotherapy [90].

### 4.2. Biochemical Markers for Ovarian Reserve

#### 4.2.1. AMH

AMH is produced by GCs from ovarian follicles, which are mainly small antral follicles. Typically, its level rapidly decreases with age and becomes undetectable during menopause [91]. Accordingly, AMH, at a low level, exhibits high sensitivity and specificity for chemotherapy-exposed women with poor ovarian reserve [92]. AMH levels before or after cancer treatment provide information regarding the diversity of gonadal damage according to the type of the chemotherapeutic agent and help predict long-term ovarian function [93]. Several studies have shown that there is an AMH level threshold that predicts the risk of amenorrhea after chemotherapy [94,95,96].

However, the level of AMH does not always correlate with the quality of oocytes because it only reflects the quantity of oocytes [97]. Additionally, AMH concentration could be altered by the handling of the blood sample or the assay method used to measure AMH levels [98].

#### 4.2.2. Basal Follicle-Stimulating Hormone and Estradiol

Follicle-stimulating hormone (FSH) concentrations usually change throughout the menstrual cycle and can be best measured during the early follicular phase. Basal FSH, which is the FSH level during the early follicular phase, is widely used to determine ovarian reserve. FSH levels >10 IU/L on menstrual cycle day 2 or 3 may indicate a decreased ovarian reserve. Elevated FSH levels in young women with amenorrhea suggest premature ovarian insufficiency.

After chemotherapy, FSH levels usually increase due to follicular depletion. However, basal FSH is not always a valuable marker of ovarian reserve in patients who have undergone cancer treatment. For example, if women have regular menstrual cycles, FSH levels may show normal values, even though the ovarian reserve decreases after treatment [89,99]. In such instances, that is, when the FSH levels are within the normal range, estradiol concentration in the early follicular phase may provide additional information [100].

#### 4.2.3. Inhibin-B

Inhibin-B is secreted by the GCs of antral follicles and it regulates FSH levels via a negative feedback reaction. Inhibin-B is usually exhibited at low levels in women with a decreased ovarian reserve [101]. However, it is not a reliable marker of the ovarian reserve because its levels vary widely during menstrual cycles [102].

### 4.3. Ultrasonographic Markers

During the early follicular phase, transvaginal ultrasound can be used to count antral follicles measuring 2–10 mm in both ovaries [103]. A low AFC may be related to a diminished response to ovarian stimulation. Furthermore, a few studies have demonstrated that a low AFC could be a marker for the risk of developing amenorrhea after cancer treatment [94,96]. However, the estimation of ovarian volume using ultrasound provides limited clinical utility as an ovarian reserve marker.

## 5. Prevention and Management of Ovarian Damage

FP options could be differentiated by patient age, desire for conception, treatment regimen, socioeconomic status, and treatment duration (Figure 4) [45]. Such options include the use of hormonal medications for ovarian suppression, various methods for protecting the ovaries during radiotherapy, cryopreservation, in vitro oocyte maturation, artificial ovaries, and stem cell technologies.

### 5.1. Prevention of Chemotherapy-Induced Ovarian Damage

With conventional chemotherapy regimens, cytotoxicity-associated ovarian insufficiency involves PF pool depletion by apoptosis or hyperactivation mechanisms, notably mediated by the ABL/TAp63 and PI3K/Akt/mTOR pathways (Figure 5).

#### 5.1.1. Gonadotropin-Releasing Hormone (GnRH) Agonists

GnRH agonists are hormonal medications that modulate gonadotropins and sex hormones [104]. Ovarian suppression via the administration of a GnRH agonist before or during chemotherapy may have protective effects on the ovaries by downregulating the secretion of FSH and pituitary luteinizing hormone [105]. Previous studies have shown that ovarian suppression with GnRH agonists during chemotherapy protects ovarian function in AYAs treated for lymphoma, breast cancer, and other diseases [106,107,108]. The American Society of Clinical Oncology guidelines state that there are conflicting recommendations regarding GnRH agonists and other means of ovarian suppression for FP [15].

GnRH analogs have two possible mechanisms of action [109,110,111]. The first involves decreasing the sensitivity of the PFs entering the growing pool to gonadotoxicity by the administration of a GnRH agonist. The second constitutes the direct anti-apoptotic effect of GnRH agonists on ovarian germline stem cells. Several studies have demonstrated that the use of GnRH agonists reduces chemotherapy-induced ovarian damage; however, the mechanism underlying the protective effect on ovaries remains unclear [112,113,114,115,116,117]. Therefore, the use of GnRH agonists is considered suitable for AYAs with cancers for whom other established FP options are not suitable—such as cryopreservation—to reduce chemotherapy-induced ovarian insufficiency. The use of GnRH agonists, in combination with other modalities, including oocyte or embryo freezing, may be a good option [118]. GnRH agonists usually do not have a protective effect against radiation treatment-induced gonadotoxicity [119,120].

#### 5.1.2. AS101

AS101 is a non-toxic immunomodulator that acts on the PI3K/PTEN/Akt pathway [121]. AS101 is known to inhibit Akt activation in mouse multiple myeloma cell lines [122]. An in vivo study demonstrated that AS101 was effective in reducing apoptosis in the GCs of growing follicles by regulating the PI3K/PTEN/Akt pathway [123]. Moreover, AS101 does not interfere with the therapeutic effect of chemotherapy and in fact shows synergistic antitumor activity [124,125,126,127]. According to an in vivo study using cyclophosphamide-treated mice, cyclophosphamide induced an increase in early growing follicles and an increased ratio of growing/dormant follicles was observed histologically. This suggests that cyclophosphamide-induced primordial follicle loss is not due to apoptosis, but the activation of primordial follicles to undergo recruitment and growth. AS101 cotreatment reduces cyclophosphamide-induced activation of PTEN/PI3K/Akt pathway proteins in the ovaries. AS101 was also found to reduce follicle loss and improve AMH concentration after cyclophosphamide treatment. In addition to preventing follicle loss and maintaining fertility, AS101 also has the added benefit of improving the responsiveness of breast cancer cells to cyclophosphamide chemotherapy [122].

#### 5.1.3. Anti-Müllerian Hormone (AMH)

AMH is produced in GCs of growing follicles and has a suppressive effect on follicle activation [128,129,130]. AMH inhibits PF recruitment and growth, and the loss of AMH results in the depletion of the PF pool. A previous study confirmed that the initiation of primordial follicle growth was inhibited when human ovarian cortical tissue was cultured with recombinant AMH [127]. The combination of recombinant AMH with the cyclophosphamide metabolite in an ex vivo culture system maintained a high number of PFs in the ovaries [24]. Moreover, when recombinant AMH was administered together in a model of CY-treated pubertal mice, FOXO3A phosphorylation, the main factor for PMF activation, was significantly reduced. The pPS6K level was decreased in the mice treated with AMH compared to the mice treated with only CY, suggesting that AMH prevents PF activation through the mTOR pathway [58]. AMH shows minimal side effects because it is an endogenous hormone with activity limited to the ovaries.

#### 5.1.4. Imatinib

Imatinib, a tyrosine kinase inhibitor used in cancer treatment, selectively inhibits the ABL kinase domain of the bcr-abl oncogenic protein present in patients with chronic myelogenous leukemia [131]. It is also known to inhibit cKIT and platelet-derived growth factor receptor tyrosine kinases in gastrointestinal stromal tumors [132,133]. As PF depletion by apoptosis or activation mechanisms are mainly mediated by the ABL/TAp63 and KIT/PI3K pathways, respectively, it can be hypothesized that imatinib might be able to prevent ovarian dysfunction caused by these pathways [134]. Previous studies have demonstrated that c-Abl helps maintain genomic integrity by managing DNA breaks [135,136]. Many studies have investigated the protective effects of imatinib, but conflicting results have been reported [137,138,139]. In PFs of postnatal day 5 mice, cisplatin induced c-Abl nuclear accumulation, increased TAp63 levels, and eventually caused cell apoptosis. Treatment with the c-Abl kinase inhibitor, imatinib, rescued cisplatin-induced depletion of the follicle reserve. The average number of pups and pregnancy rates were also higher in mice transplanted with ovarian tissue treated with combined cisplatin and imatinib [130]. However, in a similar mouse model study, the number of PFs and the average number of pups were not significantly higher in the group treated with imatinib in addition to cisplatin [133]. This conflicting result is thought to be due to the different types of cisplatin used in each study, and thus the toxicity to PFs at the same dose was different.

#### 5.1.5. Sphingosine-1-Phosphate

Sphingosine-1-phosphate (S1P), a naturally occurring sphingolipid, inhibits the ceramide-promoted apoptotic pathway [140]. It increases vascularity and angiogenesis, and reduces PF apoptosis [69]. In mice xenotransplanted with human ovarian cortical tissue, the co-administration of S1P with cyclophosphamide and doxorubicin was associated with a lower rate of apoptosis. The authors analyzed the expression of activated caspase 3 using immunohistochemistry to evaluate the activation of apoptotic cell death pathways. The expression of activated caspase 3 in human ovarian tissue cotreated with S1P was significantly lower than that in the group treated with cyclophosphamide or doxorubicin alone [141]. It also showed a protective effect in mice treated with dacarbazine [142]. However, another study found no reduction in chemotherapy-induced follicle loss in S1P-treated rats [143].

#### 5.1.6. Granulocyte Colony-Stimulating Factor

Granulocyte colony-stimulating factor (G-CSF) treatment significantly decreases the loss of PFs induced by chemotherapeutic agents, prevents damage to vascular structures, and reduces DNA damage in oocytes or growing follicles. In 6-week-old mice treated with high-dose alkylating chemotherapy (cyclophosphamide and busulfan), G-CSF with or without stem cell factor maintained PF numbers. G-CSF treatment with or without stem cell factor treatment decreases chemotherapy-induced gamma H2AX phosphorylation in early growing follicles, which is the earliest cellular response to DNA damage. G-CSF coadministration also increases microvessel density, as assessed by immunofluorescent staining for PECAM1/CD31 in vascular endothelial cells after chemotherapy treatment [69,144]. It also shows an anti-apoptotic effect; however, there is a concern that G-CSF may induce follicle loss associated with focal ischemia, fibrosis, and infarcts of blood vessels [69,145].

### 5.2. Prevention of Ovarian Damage before and during Radiation Treatment

#### 5.2.1. Identifying the Location of the Ovaries

Technical improvements in radiotherapy have maximized target coverage while sparing normal organs using highly conformal dose distributions [73]. Identifying the location of the ovaries during treatment is essential for determining the dose administered to the ovaries during radiotherapy. In one study, ovaries were identified in 75% of the patients via an abdominal computed tomography (CT) scan [146]. However, the position of the ovary is affected by the contents of the bladder. Variations in the ovary position can result in their displacement to a high-dose irradiation field [147]. Therefore, daily on-board CT imaging may be used to map the ovary position before radiation treatment. Three-dimensional imaging provides relevant information for use in adaptive radiotherapy. Additionally, patient education regarding bladder filling before treatment would help not only to maintain an accurate radiation field for effective treatment but also to improve the protection of the ovaries [73,148].

#### 5.2.2. Alternative Techniques for Craniospinal Radiotherapy

Craniospinal irradiation can induce gonadal damage [149,150]. Traditionally, craniospinal irradiation has been conducted using the standard posterior–anterior inferior spinal field. However, opposed lateral fields for the inferior spine help to reduce the radiation dose around the ovaries [151]. In other studies, proton therapy for craniospinal irradiation generated with a posterior–anterior proton beam stopping just after the thecal sac showed a reduced ovarian dose without an unexpected exit dose [152,153,154].

#### 5.2.3. Ovarian Transposition

In certain situations, the ovaries may be located proximally to the lesion targeted by the high-dose radiation. The transposition of ovaries out of the fields, which is a procedure called oophoropexy, may help to protect the ovaries against radiation-induced damage [155,156,157], and it has been recommended for all AYAs requiring radiation [158,159]. Ovarian transposition has been recommended for patients with advanced cervical cancer who require radiotherapy, but only a quarter of patients younger than 40 years have undergone oophoropexy before treatment [160,161,162,163]. Laparoscopic techniques can be used to minimize surgical morbidity [164,165]. Ovaries are usually fixed to the anterolateral abdominal wall. During the procedure, the ovarian vessels should be carefully mobilized to avoid damage. Metallic clips are usually applied at the base of the transposed ovary [159,162]. As there is a risk of gonadal failure after pelvic irradiation following ovarian transposition, concurrent ovarian tissue cryopreservation during surgery is strongly recommended [166]. External beam radiation could be the main risk factor for ovarian failure even after transposition, and combined chemotherapy may aggravate the risk [74,162,167]. Ovarian transposition should be offered to all women who have a high possibility of radiation exposure near the ovaries. However, it should not be offered to women with poor ovarian reserve, with a high risk of ovarian metastasis, or who are scheduled for chemotherapy only.

### 5.3. Cryopreservation

#### 5.3.1. Embryo Cryopreservation

Although ovarian damage can be reduced using the above-mentioned surgical or imaging-guided methods or via the administration of protective agents before chemotherapy, embryo cryopreservation is the most well-established method for preserving fertility [168]. Embryo freezing should initially be considered for FP treatment if there is adequate time for ovarian stimulation, and a partner or donor sperm is available [169]. This technique is safe and effective for patients undergoing assisted reproductive procedures [170,171]. Slow freezing and vitrification methods offer excellent results and are widely used [172]. Several studies have suggested that embryo vitrification and thawing methods are better than slow freezing and thawing methods in terms of pregnancy and live birth rates [173,174,175]. Many institutions prefer vitrification as a simpler and less expensive alternative to slow freezing, which requires the use of controlled-rate freezers and large quantities of liquid nitrogen.

Embryo cryopreservation requires ovarian stimulation; therefore, this option is not adequate for prepubertal girls. Additionally, embryo cryopreservation may not be appropriate for women who do not have a partner or do not want to use donor sperm. Furthermore, this technique may entail the risk of losing reproductive autonomy and present possible issues with the ownership of stored embryos. In patients with hormone-dependent breast cancer, aromatase inhibitors can be used as alternatives to ovarian stimulation [176]. If there is insufficient time for FP procedures before cancer treatment, random-start ovarian stimulation would be appropriate [177,178,179]. In studies comparing the results of in vitro fertilization and embryo cryopreservation in patients with cancer and those without cancer, contradictory results were observed in terms of fertilization and live birth rates [178,180,181,182].

#### 5.3.2. Oocyte Cryopreservation

Oocyte cryopreservation is another widely used method, which is considered a standard technique for FP in patients with cancer [183]. The introduction of vitrification into assisted reproductive techniques has resulted in oocyte cryopreservation outcomes similar to those obtained with fresh oocytes [184,185]. In 2013, the American Society for Reproductive Medicine (ASRM) approved oocyte cryopreservation for FP based on the results of four clinical trials [186,187,188,189]. This technique has similar disadvantages to those of embryo cryopreservation because it involves ovarian stimulation, which makes it unsuitable for prepubertal girls. However, it can be utilized for women who are single or do not want sperm donation. Vitrification was more effective than slow freezing in avoiding crystallization because the former reduced cellular damage and chilling injury during the freezing process [189,190]. Accordingly, the National Institute for Health and Care Excellence guidelines recommend vitrification instead of controlled-rate freezing for oocyte cryopreservation, given the availability of the necessary equipment and expertise [191].

As oocyte freezing involves the removal of cumulus cells before cryopreservation, it can induce changes in the zona pellucida, which may affect the fertilization rates of conventional insemination. Therefore, ASRM recommends intracytoplasmic sperm injection for frozen oocytes as the preferred procedure [168,192].

The combination of oocyte cryopreservation and ovarian tissue cryopreservation can enhance the results of the FP procedure [193]. However, the cryopreservation of ovarian tissue concomitant with oocyte retrieval is ineffective; thus, it is not recommended after ovarian stimulation with human menopausal gonadotropin or recombinant FSH followed by human chorionic gonadotropin [194,195].

#### 5.3.3. Ovarian Tissue Cryopreservation and Transplantation

Ovarian tissue cryopreservation is generally the only option for FP in children or AYAs with cancer who need immediate treatment and do not have enough time for ovarian stimulation and other procedures. Using this technique, a large number of oocytes, including PFs, can be preserved, and hormonal function of the ovary can be protected to improve the quality of life of the young patients [196].

Ovarian tissues are excised via biopsy, partial oophorectomy, or total oophorectomy, and are frozen for preservation before the initiation of cancer treatment [197]. To date, slow freezing has been established as the preferred method for ovarian tissue cryopreservation over vitrification [198]. However, several recent studies have shown promising results after vitrification [199,200,201]. A review of 60 cases showed that ovarian activity was restored in 92.9% of the cases after the transplantation of the ovarian tissue that was cryopreserved using the slow-freezing method [195]. After ovarian tissue transplantation, recovery of ovarian function has been reported, along with successful live births [183]. 

Oocyte cryopreservation is not suitable for patients with ovarian or hematologic malignancies because of the possible contamination of the ovarian tissue with malignant cells, as shown in several studies [202,203]. Nonetheless, ovarian tissue cryopreservation may be considered after an initial dose of chemotherapy to reduce the risk of malignant cell contamination, despite possible partial ovarian damage [204].

Recently, a study evaluated the changes in telomere length and senescence markers during ovarian tissue cryopreservation. The mean telomere length was significantly decreased after cryopreservation, and Western blot analysis indicated that senescence markers were affected by cryopreservation [205]. The possibility of irreversible DNA changes, such as the shortening of telomere length and alterations in senescence markers, should be considered when using this technique.

## 6. Future Perspectives on FP

### 6.1. Whole Ovarian Transplantation

Whole ovarian transplantation has the benefit of immediate revascularization following blood vessel anastomosis, thereby reducing ischemic injury; however, potential injury due to hypothermic damage to blood vessels and difficulties in dispersing a sufficient amount of cryoprotective agent make it challenging in clinical practice [206,207]. Successful whole ovarian cryopreservation and transplantation have been reported in animal studies, and whether vitrification or slow freezing is preferable for cryopreservation is still under debate [208,209,210,211]. Before whole ovarian transplantation can be established in clinical use, there may be several challenges to its adoption, including proficient and diligent use of surgical techniques and identifying optimal protocols for cryopreservation and thawing. To date, there has been no experimental model to explore techniques for the cryopreservation and retransplantation of human ovaries. Table 2 shows several studies that evaluated the feasibility of the xenotransplantation of human ovaries into animals for study purposes.

### 6.2. In Vitro Maturation

In vitro maturation (IVM) is usually used in patients with polycystic ovarian syndrome (PCOS). This requires immature oocyte retrieval and cryopreservation either at an immature stage or at a post-IVM mature state [242]. This can also be used in patients with cancer who lack adequate time for ovarian stimulation or prepubertal girls who need immediate treatment. Although many researchers have attempted to improve outcomes by combining IVM and vitrification, only a few live births have been reported after IVM procedures in patients with cancer [243,244,245]. Further technical improvements may allow the use of this technology for FP in the near future.

### 6.3. Artificial Ovaries

The transplantation of ovarian tissue with a scaffold accelerates tissue vascularization via the release of multiple bioactive materials [246,247]. Recently, this technique has been most commonly used to enable the transplantation of various substances or accessory cells, including stem cells, together with ovarian tissues [248,249,250]. Artificial ovaries can be useful for developing mature oocytes via multiple processes, including in vitro culture of oocytes, isolated follicles, and ovarian tissue [251,252]. In animal studies, this approach restored endocrine function, enabling in vivo follicle development and successful pregnancy; however, there have been no successful reports in humans [252,253]. Improvements in techniques and addressing genetic safety concerns are warranted for more consistent results in human applications.

### 6.4. Ovarian Stem Cells

Stem cells are being investigated for use in FP. One study reported the successful detection and isolation of ovarian stem cells in animals and humans [254,255]. Additionally, egg-producing stem cells isolated from ovaries differentiate into young oocytes [254]. This technique may provide another option for prepubertal children with cancer and women for whom conventional FP methods are not suitable. However, it is not commonly applied in clinical practice because of insufficient evidence in human-assisted reproduction, the scarcity of ovarian stem cells, and the ethical issues related to the use of oocytes and embryos [256]. Further studies are required to implement these approaches in clinical practice.

## 7. Conclusions

Improving our knowledge regarding the molecular mechanisms involved in cancer therapy-induced ovarian damage can lead to the development of treatments to limit follicular damage. Some of the molecular mechanisms involved in the protective effects of various agents are unclear. Although several studies have assessed the effect of disease and treatment on human ovaries, few studies have been conducted because of ethical concerns [257,258]. However, understanding the molecular etiology of treatment-induced ovarian dysfunction can not only aid in identifying targets to prevent and reduce gonadal damage during cancer treatment, but also increase the number of options for FP.

Gonadotoxic cancer treatments induce iatrogenic premature ovarian insufficiency and loss of fertility in prepubertal girls and AYAs with cancer. An individualized strategy including established and experimental techniques should be provided based on patient age, marital status, economic status, chemotherapy regimen, cancer type, staging upon diagnosis, and available time for the FP process to prevent loss of ovarian function and fertility.

Novel therapies are of great interest, as they may limit follicular loss and protect the ovaries before or during cancer treatment. These therapies could be utilized in combination with standard FP techniques, or they may be used alone in the future. These strategies can assist young women who are not eligible for traditional FP methods because of their age or limited time before the initiation of disease treatment. Effective multidisciplinary oncofertility strategies involving a highly skilled and experienced team composed of medical oncologists, gynecologists, reproductive biologists, surgeons, patient care coordinators, psychologists, and research scientists should be carefully considered for each patient to provide high-quality care.

## Figures and Tables

**Figure 1 ijms-22-07484-f001:**
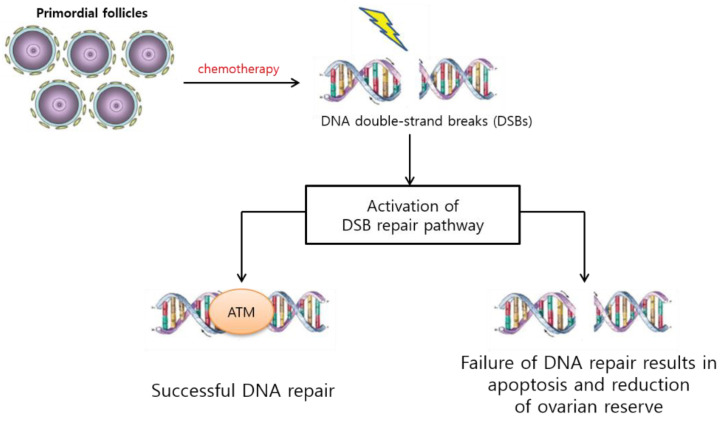
Chemotherapy-induced DNA double-stranded breaks (DSBs). If the DSB is repaired successfully, the cell survives via ataxia–telangiectasia mutated (ATM)-mediated DNA damage repair pathways. However, DNA damage can result in cellular apoptosis and a reduction in the ovarian reserve if the repair pathway fails.

**Figure 2 ijms-22-07484-f002:**
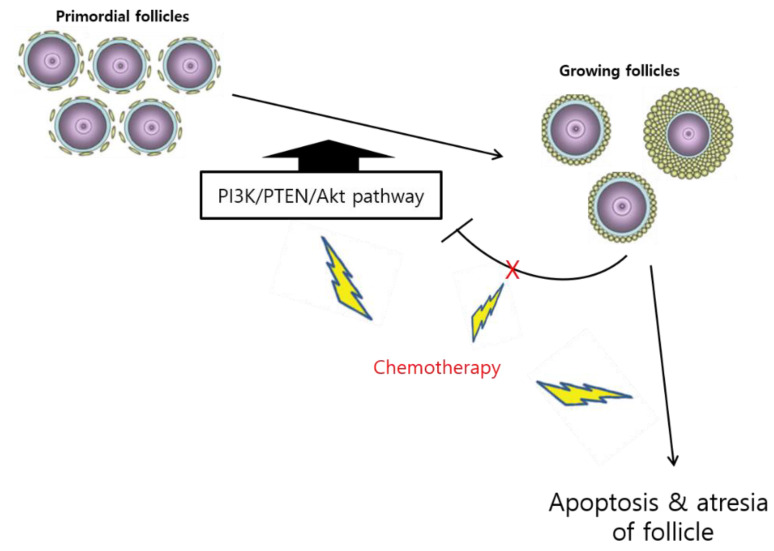
“Burnout” theory of ovarian follicle reserve. The chemotherapy-induced PI3K/PTEN/Akt pathway activates the destruction of follicles, followed by an impairment of AMH and the loss of the suppression of the primordial follicle (PF) pool. Consequently, the PFs are activated to compensate for the decrease in the number of growing follicles.

**Figure 3 ijms-22-07484-f003:**
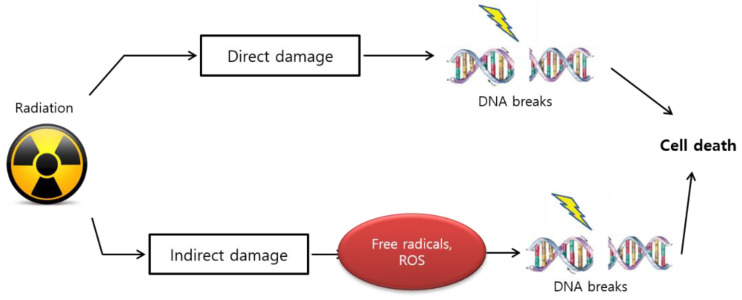
Biological effect of radiation via linear energy transfer (LET) in tumors. High LET radiation results in “direct” gonadal DNA damage that incorporates multiple lesions within the helical turns of the DNA molecule. Conversely, the increase in reactive oxygen species (ROS) induces rapid primordial follicle loss via “indirect” damage.

**Figure 4 ijms-22-07484-f004:**
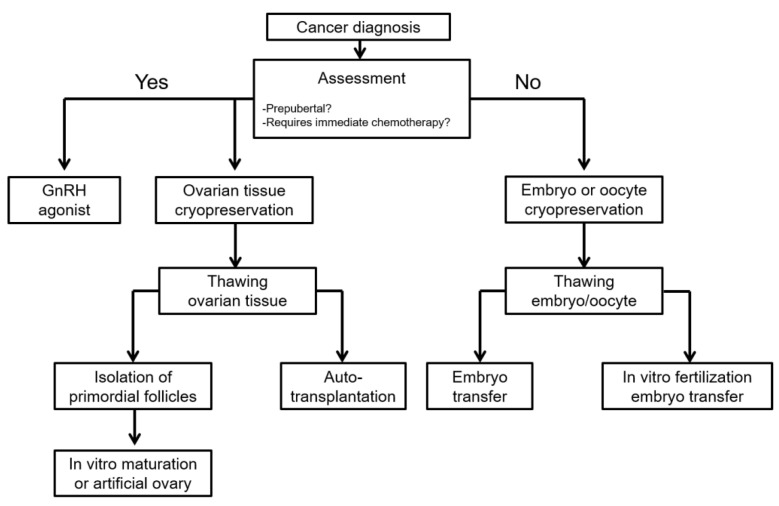
Fertility preservation options for female cancer patients. This could be differentiated by individual situations. Adapted and reprinted, with permission, from Cho, 2020 [45].

**Figure 5 ijms-22-07484-f005:**
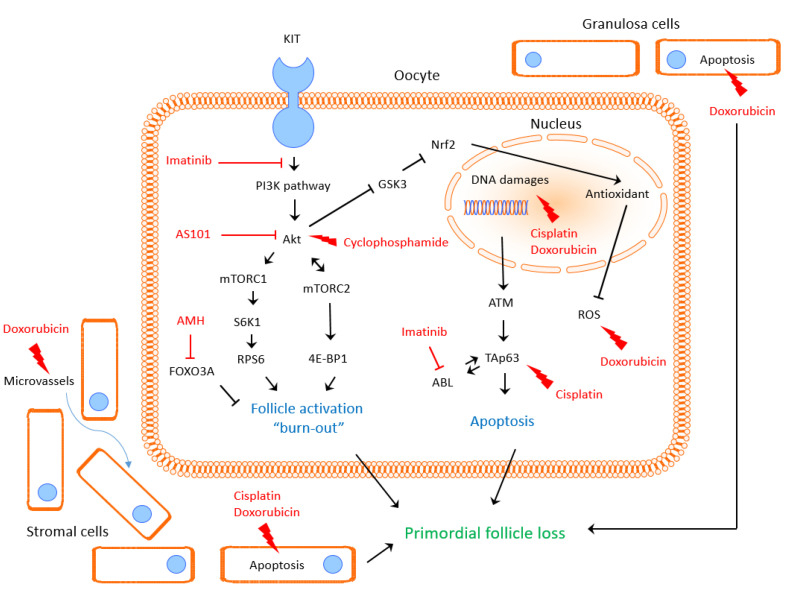
Impact of cytotoxic agents and protective molecules on pathways involved in primordial follicle pool loss.

**Table 1 ijms-22-07484-t001:** Ovarian damage risk with chemotherapeutic agents and their respective mechanisms of action.

Class	Agents	Type of Cancer	Mechanisms of Action	Damage Risk
Alkylating agents	Cyclophosphamide Ifosfamide Nitrosureas Chlorambucil Melphalan Busulphan Mechlorethamine	Leukemia, breast cancer, lung cancer, ovarian cancer, prostate cancer, lymphoma, myeloma, sarcoma, Hodgkin’s disease	Interference with cell division via intra-strand/inter-strand cross-linking of DNA. Induction of a reduction in mitochondrial transmembrane potential. Inhibition of the accumulation of cytochrome c in the cytosol. Induction of DSBs in oocytes and GCs.	High
Vinka alkaloids	Vinblastine Vincristine	Testicular cancer, lymphoma, Hodgkin’s disease, breast cancer, germ cell tumors, lung cancer, sarcoma, neuroblastoma	Inhibition of tubulin from forming into microtubules. Not gonadotoxic.	Low
Antimetabolites	Cytarabine Methotrexate 5-fluorouracil	Leukemia, breast cancer, ovarian cancer, gastrointestinal cancer	Inhibition of purine, pyrimidine becoming incorporated into DNA during S phase of cell cycle. Inhibition of RNA synthesis. Not gonadotoxic.	Low
Platinum agents	Cisplatin Carboplatin Oxaliplatin	Bladder cancer, colorectal cancer, head and neck cancer, lung cancer, ovarian cancer, testicular cancer	DNA damage by formation of inter-strand/intra-strand DNA adducts which interfere with cellular transcription and replication. Accumulation of abl and TAp63-alpha protein in the oocyte leading to oocyte death.	Moderate
Anthracycline antibiotics	Daunorubicin Bleomycin Doxorubicin	Lymphoma, leukemia, breast cancer, sarcoma	Intercalation with DNA and prevention of its replication and transcription via the inhibition of topoisomerase II. Upregulation of P53 protein which induces apoptosis in the presence of high levels of DNA damage. DNA DSBs leading to activation of ATM, which initiates apoptosis. GCs are usually targeted due to their mitotically and metabolically active characteristics.	Low/moderate
Others	Procarbazine	Hodgkin’s disease, brain cancer	Inhibition of DNA methylation, and RNA and protein synthesis.	High

**Table 2 ijms-22-07484-t002:** Experiments which evaluated the feasibility of human ovarian xenotransplantation into animals.

Evaluation	Animal	Materials	Transplantation Site	Publication
Oocyte maturation after xenotransplantation	SCID mice	Ovarian tissue	Back muscle, kidney capsule	Soleimani et al. [212]
	SCID mice	Ovarian tissue	Kidney capsule	Gook et al. [213,214]
	SCID mice	Ovarian tissue	Neck muscle	Lotz et al. [215]
	SCID mice	Ovarian tissue	Subcutaneous space	Kim et al. [216]
Follicular development	SCID mice	Ovarian tissue	Intraperitoneum	Ayuandari et al. [217]
	NOG mice	Ovarian cortex	Back skin, kidney capsule, ovarian bursa	Terada et al. [218]
	SCID mice	Clot containing preantral follicle	Ovarian bursa	Dolmans et al. [219]
	SCID/hpg mice	Ovarian tissue	Kidney capsule	Oktay et al. [220]
	Swiss nu/nu mice	Ovarian tissue	Intraperitoneum	David et al. [221]
	NOD/SCID mice	Ovarian tissue	Subcutaneous space	Weissman et al. [222]
	NMRI nu/nu	Ovarian tissue	Intraperitoneum, subcutaneous space, ovarian bursa, thigh muscle	Dath et al. [223]
	Swiss nu/nu mice	Clot containing preantral follicle	Intraperitoneum	Paulini et al. [224]
	SCID mice	Ovarian tissue	Intraperitoneum	Amorim et al. [225]
	NMRI nu/nu	Ovarian tissue	Thigh muscle	Jafarabadi et al. [226]
	NOD/SCID mice	Ovarian cortex	Subcutaneous space	Campos-Junior et al. [227]
Optimization of grafting protocols	NMRI nu/nu	Ovarian tissue	Intraperitoneum	Van Eyck et al. [228]
	SCID mice	Ovarian cortex	Back muscle	Soleimani et al. [69]
	B6cg nude mice	Ovarian tissue	Back skin, thigh muscle	Hormozi et al. [229]
	NOD/SCID mice	Ovarian cortex	Kidney capsule, subcutaneous space	Hernandez-Fonseca et al. [230]
	SCID mice	Ovarian tissue	Neck muscle	Maltaris et al. [231]
	Balb/C nu/nu	Ovarian tissue	Back muscle	Friedman et al. [232]
	SCID mice	Ovarian stroma	Subcutaneous space	Fu et al. [233]
	NMRI nu/nu	Ovarian tissue	Intraperitoneum	Van Langendonckt et al. [234]
	New Zealand rabbits	Ovarian tissue	Back muscle	Wang et al. [235]
	Balb/C nu/nu	Ovarian tissue	Ovarian bursa, subcutaneous space	Ruan et al. [236]
Reimplantation of malignant cells	SCID mice	Ovarian cortex	Intraperitoneum	Luyckx et al. [237]
	NOD/SCID mice	Ovarian tissue	Subcutaneous space	Kim et al. [238]
	NMRI nu/nu	Ovarian tissue	Subcutaneous space	Greve et al. [239]
	SCID mice	Ovarian tissue	Neck muscle	Lotz et al. [240]
	SCID mice	Ovarian tissue	Intraperitoneum	Dolmans et al. [241]

NMRI: National Medical Research Institute; NOD: non-obese diabetic/severe combined immunodeficient; NOG: non-obese diabetic/severe combined immunodeficient/γcnull; SCID: severe combined immunodeficient; SCID/hpg: severe combined immunodeficient/hypogonadism.

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
