# Peer review of "Molecular Mechanism and Prevention Strategy of Chemotherapy- and Radiotherapy-Induced Ovarian Damage"

_ijms, 2021, doi:10.3390/ijms22147484_

Round 1
Reviewer 1 Report
The authors have addressed my previous comments and I think the manuscript is acceptable in the journal.
Reviewer 2 Report
All my previous comments are addressed and I do not have any further comments.
This manuscript is a resubmission of an earlier submission. The following is a list of the peer review reports and author responses from that submission.
Round 1
Reviewer 1 Report
Please find my comments in the attachment.
regards

Reviewer 2 Report
The authors have prepared a very nice and extensive overview on cancer treatment induced ovarian dysfunction and fertility preservation.
Please find below some minor suggestions for improvements;
section 4.2.3 (line 276). It could be considered to add a sentence stating that Ovarian transposition should not be offered to women scheduled to undergo chemotherapy, with reduced ovarian reserve or high risk of ovarian metastasis.
Section 4.3.1. Line 290 - please adapt this sentence. This technique required OS and therefore this option is not adequate for prepubertal girls. Also, embryo cryo may not be appropriate in women that do not have a partner or do not want to use donor sperm (the latter is not due to OS).
Furthermore, a sentence could be added that embryo cryo entails the risk of losing reproductive autonomy and possible issues with ownership of stored embryos.
Line 294 - random start can be considered, but this is not to prevent OHSS. This should be corrected.
Line 296 - I consider there are contradictory results with regards to the success rates of embryo cryo in women with cancer and other patients. It is suggested to make the statement less strong.
Line 319: this sentence may need revision.
Section 4.3.3. line 324 - please revise this sentence. OTC is generally the only option for children, and indeed for urgent FP in AYAs, but this is not due to OTC being able to be performed at any stage of the menstrual cycle.
line 362 - lacking adequate time for Ovarian stimulation (not FP)
line 364 - there have been a few reports of live births (Prasath, et al., 2014), (Uzelac, et al., 2015)
Reviewer 3 Report
In this review, the authors have gathered information about the molecular mechanisms for treatment-mediated ovarian dysfunction and infertility in cancer patients. This is an interesting and important review and will be an informative source for the readership including oncologists and cancer biologists. To this reviewer, there are a few minor points that need to be addressed by the authors:
1-Human genes and proteins should be capital throughout the manuscript.
2-Table 1: Because the manuscript is about the treatment-induced ovarian malfunction, it may be more informative if the table shows different classes of chemotherapies (alkylating agents, platinum-based chemotherapies, etc.), examples of each class, the type of cancer they are prescribed for, their mechanism of action and finally, their infertility/damage risk.
3-Could the authors discuss the clinical detection of ovarian damage after chemotherapy? This could be one paragraph and includes clinical, biochemical, ultrasonographic markers, etc.
Round 2
Reviewer 1 Report
I appreciate the revised work of authors but unfortunately my major comments regarding similarity with other published articles were not addressed during revision.